# Effects of Sous Vide Cooking on the Physicochemical and Volatile Flavor Properties of Half-Shell Scallop (*Chlamys farreri*) during Chilled Storage

**DOI:** 10.3390/foods11233928

**Published:** 2022-12-05

**Authors:** Yuexiang Zhan, Chuanhai Tu, Huili Jiang, Soottawat Benjakul, Jilong Ni, Kaixuan Dong, Bin Zhang

**Affiliations:** 1Pisa Marine Graduate School, Zhejiang Ocean University, Zhoushan 316022, China; 2Key Laboratory of Health Risk Factors for Seafood of Zhejiang Province, College of Food Science and Pharmacy, Zhejiang Ocean University, Zhoushan 316022, China; 3International Center of Excellence in Seafood Science and Innovation, Faculty of Agro-Industry, Prince of Songkla University, Hat Yai 90110, Thailand

**Keywords:** sous vide, scallop, muscle quality, volatile compounds, chilled storage

## Abstract

This study explored the effects of sous vide (SV) cooking treatments on the physicochemical quality and volatile flavor of half-shell scallop (*Chlamys farreri*) during 30 d of chilled storage. The vacuum-packed scallop samples were cooked at 70 °C (SV-70) and 75 °C (SV-75) and maintained for 30 min. The samples were compared with the positive control (cooked at 100 °C for 10 min, CK). The results indicate that the total volatile basic nitrogen (TVBN), pH, texture, and malondialdehyde (MDA) content gradually increased, while the myofibrillar protein (MP) extraction rate of the CK, SV-70, and SV-75 samples significantly decreased with increasing chilled storage time. Significantly, the SV cooking treatments maintained a much higher water-holding capacity of scallop muscle, compared with the conventional cooking process at 100 °C. Additionally, the SV-75 cooking treatment maintained relatively stable TVBN, pH, and MDA content, springiness, and shearing force properties of scallop samples, especially during 0–20 d of storage. Volatile flavor analysis showed that a total of 42 volatile organic compounds (VOCs) were detected in the scallop samples, and there were no considerable differences in these VOCs between the CK and SV-75 cooked samples (0 d). Overall, the SV cooking treatments effectively maintained acceptable and stable physicochemical and volatile flavor properties of half-shell scallop samples during chilled storage.

## 1. Introduction

Scallop (*Chlamys farreri*) is very popular among consumers because of its relatively large size, fast growth, rich nutrition, and delicious taste. According to a report by the China Fishery Statistics Yearbook [1], the mariculture production of scallops in 2020 reached 1.828 million tons, and scallops were mainly produced in Shandong, Liaoning, and Guangdong Provinces of China. Due to their high moisture and protein content, fresh bivalve molluscs such as mussels and scallops are highly perishable after death, and their rapid deterioration is mainly caused by the work of endogenous enzymes and microorganisms during storage [2,3]. Therefore, scallops are commonly processed in the form of dried, cooked, and/or frozen products [4,5]. Dried scallop products have higher requirements for their storage conditions, and the meat is very rough and hard. Refrigerated scallop products are also susceptible to deterioration via protein denaturation, oxidation, bacterial invasion, and protease degradation during long-term cold storage. The traditional 100 °C heating method is the most common processing method, but it induces significant drip loss of nutrients and water, decreases tenderness, and produces an overripe taste in the scallop products [6]. The shortcomings of this traditional cooking method are well known, and further research is required to improve the cooking process for household consumption and industrial production.

Sous vide (SV), a new type of processing method, is a controlled cooking method of vacuum-packaged food, using water or steam as the heating medium under suitable temperatures and time conditions [7]. It is widely used for cooking foie gras and beef in European countries. In recent years, it has been applied to aquatic products to improve their taste and nutritional value, such as crab lump meat [vii], large yellow croaker [8], and European sea bass [9]. Significantly, this low-temperature heating treatment reduces the loss of nutritional compounds and inactivates enzymes and microorganisms in the products. Moreover, the SV-cooked products are vacuum-packed, which helps to prevent exogenous bacteria contamination, oxidation reactions, and the presence of other contaminants during storage. Zavadlav et al. [10] showed that SV-cooked salmon could be stored at 4 °C for up to 24 d, and the shelf life of SV-cooked cod was extended to 32 d under chilled conditions.

In the case of scallop products, consumers usually prefer to use their shells as containers when the scallop products are processed, cooked, and/or consumed. However, there are few studies on SV-processed bivalve molluscs, especially for half-shell scallop products. There is no large-scale industrial preparation and production of ready to eat half-shell scallops. In this study, half-shell scallops treated with different cooking conditions were evaluated during storage. We aimed to explore the changes in the physicochemical and volatile flavor properties of scallop products. This work provides theoretical support for the implementation of industrial-scale SV preparation of marine bivalve molluscs, such as scallops, mussels, and clams.

## 2. Materials and Methods

### 2.1. Chemical Reagents

Boric acid, sodium chloride, absolute ethanol, acetic acid, disodium hydrogen phosphate, trichloroacetic acid (TCA), and sodium dihydrogen phosphate were obtained from Sinopharm Chemical Reagent Co., Ltd. (Suzhou, China). Magnesium oxide was purchased from Aladdin Biochemical Technology Co., Ltd. (Shanghai, China). A malondialdehyde (MDA) assay kit was obtained from the Nanjing Jiancheng Institute of Biological Engineering (Nanjing, China). All chemicals used in this study were of analytical grade.

### 2.2. Scallop Samples and Treatments

Live scallop (*Chlamys farreri*) samples (an average weight of 65 ± 5 g, an average length of 6.3 ± 0.5 cm, and an average height of 3.0 ± 0.4 cm; 150 individuals) were obtained from a local aquatic product market in Zhoushan (Zhejiang Province, China; harvested in May 2021), and were packed in an insulated box filled with crushed ice and transported to the lab within one hour. All scallop samples were cleaned with tap water to remove the sand and dirt on the surface of the shell. Next, the adductor muscle was removed from one side of the shell using a sterile scalpel, and the entire scallop muscle was contained within a half-shell. The mantle and viscera were manually removed. Subsequently, the obtained half-shell scallops were packaged independently in a heat-resistant polyethylene (PE) bag (12 × 17 cm; obtained from Hebei Wangshi Packaging Co., Ltd., Shijiazhuang, China). The samples were vacuum-packed in a vacuum sealer (DZ-400-2S, Xinqi Machinery Group Co., Ltd. Shanghai, China). The sealed half-shell scallops were randomly divided into three groups, namely an SV-70 group (immersed in a water bath at 70 °C for 30 min; 50 individuals), an SV-75 group (immersed in a water bath at 75 °C for 30 min; 50 individuals), and a positive control group (CK; immersed in a water bath at 100 °C for 10 min; 50 individuals). Next, the cooked samples were rapidly cooled in an ice water bath for 10 min with continuous stirring. Finally, the obtained samples were stored in a refrigerator at 4 °C for 30 d, and the samples were removed and measured every 5 d.

### 2.3. Total Volatile Basic Nitrogen (TVBN) Content and pH Analysis

TVBN content in the scallop adductor muscle was determined via a method reported by Han et al. [11]. The results were calculated as milligrams N/100 g muscle. The pH value was determined by dipping a pH electrode (E-201F, INESA Scientific Instrument Co., Ltd., Shanghai, China) into the homogenate of the adductor muscle in 0.85% NaCl solution.

### 2.4. Weight Loss Analysis

After removing the water on the surface of the shell and muscle, the half-shell scallop samples were weighed via a method reported by Ortuño et al. [12]. Weight loss was calculated according to the difference in weight before (M_1_) and after the storage (M_2_), according to the following formula:Weight loss (%) = (M_2_ − M_1_)/M_1_ × 100.(1)

### 2.5. Texture Analysis

The springiness of the muscle samples was measured using a TMS-Pilot texture analyzer (TMS-Pilot, FTC, VA, USA) coupled with a 50 mm diameter cylindrical probe, according to the method reported by Zhu et al. [13]. The testing was performed in two consecutive cycles at 30% compression, a constant speed of 1 mm/s, and a trigger point of 0.06 N. Each assay of each sample was repeated six times. The shearing force of the muscle samples was determined by using a Warner–Bratzler blade at a constant speed of 1 mm/s. The muscle samples were sheared perpendicularly along the fiber direction [14].

### 2.6. Malondialdehyde Content Analysis

MDA content in the scallop muscle was measured using an MDA assay kit according to a method described by Zhu et al. [15]. The homogenate (10,000 r/min for 60 s; by using a GM200 tissue grinder (Shanghai Instruments and Equipment Co., Ltd., Shanghai, China)) of muscle samples was combined with trichloroacetic acid (TCA) solution and mixed with thiobarbituric acid (TBA). Next, the obtained mixture was heated in boiling water for 40 min and then cooled with cold water to room temperature. After centrifugation at 4000× *g* for 10 min, the absorbance of the supernatant was measured at 532 nm using a U-2600 UV-Vis spectrophotometer (Shimadzu (China) Co., Ltd., Shanghai, China). The results were expressed as mg MDA/kg of muscle.

### 2.7. Myofibrillar Protein (MP) Extraction Rate Analysis

The scallop muscle was mixed with 5 volumes of phosphate buffer solutions (50 mmol/L, pH 7.2) and homogenized for 60 s in a T18 Ultra-Turrax homogenizer (IKA, Baden-Württemberg, Germany). The homogenate (10,000 r/min for 60 s) was centrifuged at 8000× *g* for 10 min at 4 °C (5424R, Eppendorf, Saxony, Germany). The obtained sediment was collected and extracted again using the same buffer solutions. Next, the pooled precipitate was further homogenized in 10 volumes of ice-cold phosphate buffer solutions (50 mmol/L, containing 0.6 mol/L NaCl, pH 7.2). Then, the mixture was centrifuged at 8000× *g* for another 10 min at 4 °C. The collected supernatant was recognized as the MP extract [16]. The content of MPs in scallop muscle was measured using the Bradford method. The MP extraction rate was calculated according to the difference in the content of MPs from the fresh sample (C_1_) and cooked samples (C_2_) with the same mass, with the formula [17]:MP extraction rate (%) = C_2_/C_1_ × 100.(2)

### 2.8. Volatile Compound Analysis

In this experiment, the fresh scallop (raw), 100 °C-cooked samples stored for 0 d (CK-0), SV-75 samples stored for 0 d (SV-75-0), 15 d (SV-75-15), and 30 d (SV-75-30) (as representatives) were selected to determine the composition of volatile organic compounds in the scallop muscle via gas chromatography (GC) coupled with a high-resolution ion mobility spectrometry (IMS) (FlavourSpec© 1H1-00128, G.A.S, Dortmund, Germany). The scallop muscle (2.00 g) was transferred into a 20 mL headspace bottle and incubated at 75 °C for 15 min. Then, 500 μL headspace gas was injected into an MXT-5 column (15 m × 0.53 mm × 1.0 μm) using nitrogen as the carrier gas [18]. The flow rate of the carrier gas was as follows: 2 mL/min maintained for 2 min, 10 mL/min maintained for 10 min, 100 mL/min maintained for 20 min, and 150 mL/min maintained for 30 min. Next, the separated compounds were ionized in the IMS ionization chamber at 45 °C [19]. The volatile compounds were identified by comparing their mass spectra with those stored in the National Institute of Standards and Technology (NIST) spectral database and the Ion Mobility Spectrometry spectral database. The changes in the peak intensity reflected the changes in the volatile compound content between different scallop samples.

### 2.9. Data Analysis

Three parallels were performed for each sample, except for the springiness and shearing force determinations (six parallel measurements). Significance analysis was executed using SPSS v26.0 (SPSS Inc., Chicago, IL, USA) at the significance level of *p* < 0.05. Data are displayed as the mean ± standard deviation (SD). Fingerprint and principal component analyses (PCAs) were performed using VOCal software (version 0.1.1, G.A.S., Dortmund, Germany).

## 3. Results and Discussion

### 3.1. TVBN and pH Analysis

TVBN content is an important indicator to characterize the degree of protein degradation, and it is also used to evaluate the freshness of muscle products [20]. As shown in Figure 1A, the TVBN content of three groups of scallop samples increased with extended storage time. The initial TVBN contents of the SV-70 and SV-75 samples were comparatively lower than that of the positive control (CK) samples. Generally, the rapid increases in the TVBN content of scallop muscle were attributed to the growth of spoilage bacteria and associated metabolites, as well as the actions of endogenous enzymes during processing and storage. During initial storage, the retarded TVBN values of SV-70 and SV-75 samples were mainly due to oxygen deficiency and heating treatments, while the TVBN values increased remarkably during the following 5–15 d of storage, suggesting that the ammonia and amine nitrogenous substances significantly accumulated in muscle tissues, resulting from the decarboxylation and deamination of amino acids, induced by microorganisms, metabolic enzymes, and protein/lipid oxidation. Although the TVBN values of the SV-70 and SV-75 samples on day 30 increased to 7.60 and 8.92 mg/100 g of muscle, respectively, they were much lower than the acceptable limit (≤30 mg/100 g), as defined by the European Commission. These results are also consistent with the previous report for European seabass (*Dicentrarchus labrax*) fillets and Atlantic salmon (*Salmo salar*) slices by Kritikos et al. [21]. Herein, the SV cooking method inhibited the degradation of muscle proteins induced by microorganisms and endogenous/exogenous enzymes to a certain extent. Moreover, the method prolonged the shelf life of scallop products, although the effect was significantly lower than that observed in the CK samples, especially during prolonged storage periods [vii]. SV will reduce the number of bacteria as it is a heat treatment. Pino-Hernández et al. [22] indicated that SV cooking (60 °C) significantly reduced the number of bacteria in pirarucu (*Arapaima gigas*) fillets. Humaid et al. [23] also illustrated that SV cooking (65 °C) extended the shelf life of lobster (*Homarus americanus*) tails mainly by inhibiting microbial activity. Additionally, these results of TVBN content and their variations are similar to those of the current study. The microbiological test of the half-shell scallops during storage is still in progress using high-throughput sequencing procedures, including a 16S rRNA analysis and an 18S rDNA analysis.

As illustrated in Figure 1B, the changes in pH values were similar to those of the TVBN content of scallop samples during 30 d of chilled storage. The results indicate that the pH values of SV-cooked scallops were relatively stable during the initial 0–15 d of storage, while significant increases were observed during the following 15–25 d (*p* < 0.05). These changes were attributed to the increased abundance of spoilage bacteria and the accumulation of alkaline substances in muscle tissues [vii]. The pH values of the SV-75 samples were similar to that of CK, but significantly (*p* < 0.05) lower than those of the SV-70 samples on day 30; it was suggested that SV cooking at 75 °C likely hindered the growth and reproduction of bacteria, as well as limited the activity of endogenous/exogenous enzymes during chilled storage [24].

### 3.2. MP Extraction Rate Analysis

The extraction rate of MPs can be used to evaluate the denaturation degree of muscle samples after thermal processing. Muscle tissues were completely denatured when the extraction rate of MPs was less than 10% [17]. As depicted in Figure 2, the MP extraction rate of CK, SV-70, and SV-75 samples were all less than 10% after the cooking treatments, indicating that the mild SV conditions were acceptable to denature the MPs in the scallop samples. Moreover, the MP extraction rate (MP denaturation) of the SV-70 and SV-75 samples was significantly (*p* < 0.05) higher than that of the CK samples during 0–30 d of storage, and this was mainly due to the application of low temperatures. It was suggested that mild SV cooking treatment comparatively maintained the MP structure and function to a certain extent and reduced the degree of MP denaturation, which was beneficial for the preservation of the texture and water-holding capacity of cooked muscle tissues during storage. The lateral shrinkage of muscle fibers commonly occurs between 45 °C and 65 °C in response to myosin denaturation, while longitudinal shrinkage is mainly associated with actin denaturation, which occurs between 70 °C and 75 °C [25]. In the current study, the SV-cooked samples at 70 °C and 75 °C showed different MP extraction rates, which were also in agreement with the different denaturation temperatures of the MPs in the muscle. The MP extraction rate of the three sample groups decreased continuously with prolonged storage, and this could be attributed to the inhibition of aerobic bacteria and endogenous enzymes during chilled storage [26]. In addition, the SV-70 samples showed a higher decrease in the MP extraction rate than that of SV-75 samples, indicating that SV-70 cooking might be less effective in limiting the activation of spoilage bacteria and endogenous enzymes, as compared to SV-75 cooking treatments.

### 3.3. Weight Loss Analysis

As shown in Figure 3, the CK group had the highest weight loss (cooking loss) during the storage period, which was mainly induced by high temperature during the boiling process. High-temperature stress caused serious denaturation and destruction of myofibrillar proteins in muscle tissues, resulting in shrinkage of muscle fibers and collagen and subsequent extrusion of water molecules from intracellular and intercellular space [27]. For the 100 °C-cooked samples, a large amount of drip loss containing some water-soluble nutrients was a clear shortcoming, which greatly affected the sensory and nutritional values of scallop muscle products. By comparison, the weight loss of SV-70 and SV-75 samples was always lower than that of the CK samples over 30 d of storage. This suggests that the SV cooking process effectively reduced the drip loss of scallop muscle and maintained a higher water-holding capacity of scallop muscle tissues during chilled storage [28]. During 10–30 d of storage, the weight loss of SV-70 samples was comparatively higher than that of the SV-75 samples, indicating a rapid decrease in quality occurred in the SV-70 samples, and this was most likely due to the growth and reproduction of spoilage bacteria over time. It was also suggested that the heating treatment at 70 °C might not be as sufficient as the 75 °C treatment to maintain the scallop muscle quality and limit the activity of bacteria during 30 days of storage. These findings are consistent with the results of previous TVBN and pH measurements, which also indicate the 75 °C treatment is better for maintaining the scallop muscle quality than the 70 °C treatment. Importantly, the SV-75 treatments showed positive effects on the water-holding capacity (juiciness characteristic) of scallop muscle tissues during chilled storage.

### 3.4. Springiness and Shearing Force Analysis

Springiness indicates the ability of muscle tissue to recover its deformation within a certain period. As shown in Table 1, the springiness of each of the three groups of scallop samples showed a decreasing trend during the storage time. This may be due to the structural alterations of the muscle fibers, as well as its connective tissues, leading to a decline in springiness. Previously, Zhao et al. [29] illustrated that the degradation of myofibrillar proteins in large yellow croaker (*Pseudosciaena crocea*) negatively affected the springiness properties of muscle tissues during storage. Significantly, the springiness values of the SV-70 and SV-75 samples were comparatively lower than those of the CK samples during storage, which were associated with the denaturation levels in different cooked scallop samples. The CK and SV-75 treatments caused thermal denaturation (e.g., cross-linking, uncoiling, and aggregation) of muscle proteins (including collagen, myofibrillar, and sarcoplasmic proteins) and shrinkage of connective tissues, which resulted in a comparative delicate texture [6]. There was no significant difference (*p* > 0.05) between the springiness of the CK and SV-75 samples, both of which were significantly better than the SV-70 samples (*p* < 0.05) during the entire storage period. Continuous deterioration of the springiness metric occurred in the SV-70 samples over 30 d of storage, which might be due to the resuscitation of anaerobic bacteria and the intrinsic biological factors involved in endogenous enzymes (e.g., calpain and cathepsin) [28].

Shearing force is related to the tenderness and juiciness of muscle tissues, and it directly affects the consumer acceptance of meat products [30]. During chilled storage, the shearing force of three groups of scallop samples increased first and then decreased, and this trend was closely related to the denaturation of myofibrillar proteins, the actions of endogenous enzymes, and/or the microbial activity and its metabolites [31]. For CK samples, treatment at 100 °C induced a large amount of weight loss (drip loss) and a decrease in the water-holding capacity of scallop muscle tissues. This led to an increase in shearing force (tougher texture) in the cooked muscle samples. By comparison, the SV-70 and SV-75 samples exhibited significantly improved shearing force (tenderness). The SV treatments with considerably milder temperatures limited the heat damage to the muscle proteins and connective tissues, as well as reduced the liquid loss and water-soluble nutrients. Thus, the textural properties were improved, compared with the conventional 100 °C cooking process [32]. Additionally, SV heating provided a uniform and efficient transfer of heat from outside to inside the muscle in a water bath and ensured the preservation of volatile flavor compounds under vacuum packaging conditions.

### 3.5. MDA Content Analysis

As a secondary product of lipid oxidation, the content of MDA is commonly used to reflect the development and degree of lipid oxidation in muscle products. The variations in MDA content in the scallop muscle samples during chilled storage are shown in Figure 4. During the initial 0–15 d, the MDA content of all samples was relatively stable. The MDA content significantly accelerated during the following 20–30 d of storage, which suggests considerable lipid oxidation in muscle tissues during storage. The heating treatments induced the dissociation of heme proteins and promoted the disruption of cell membranes and the liberation of free iron. These changes advanced lipid oxidation in muscle tissues during chilled storage [33]. As heating temperature and time increased, the development of lipid oxidation in the muscle samples also increased, which in turn caused undesirable changes in the texture, water-holding capacity, and nutritive values [34]. In the current study, the mild SV treatments effectively hindered the lipid oxidation reaction under the limited oxygen conditions during storage, compared with the CK samples. Similar observations were also found in largemouth bass (*Micropterus salmoides*) [35] and Segureño lamb meat [12], in which samples were pretreated with the SV cooking treatments. Díaz et al. [36] determined that the MDA value of SV-processed salmon was maintained at 2.30 mg MDA/kg after 10 weeks of frozen storage, which was not sufficient to detect rancidity in the salmon products. Bongiorno et al. [37] indicated that SV cooking at 85 °C for 10 min with salt brine resulted in being able to maintain the quality of mussels and extend their shelf life to 21 days under chilled conditions, while the mussels cooked traditionally (at 90 °C for 10 min) showed a shelf life of about 14 days. These findings suggest that the SV cooking method significantly improved the oxidative stability of lipid species, compared to the traditional cooking method.

### 3.6. Volatile Organic Compound (VOC) Analysis

The changes in volatile organic compounds (VOCs) were investigated in the SV-75 samples (as representatives; SV-75-treated samples presented better stability than the other samples) during chilled storage (Figure 5). The GC-IMS spectrum of the fresh scallop (raw) was used as a reference, and it was deducted from the other GC-IMS spectra of the scallop samples. In the plot (Figure 5A), the white color represents the same signal strength of the VOCs; the blue color represents the lower signal strength of the VOCs; and the red color represents the higher signal strength of the VOCs in the samples, compared with the fresh scallop (raw). The results show considerable differences in the VOCs between the fresh and cooked scallop samples, with no apparent differences between the CK-0 and SV-75-0 samples. This indicates that the SV cooking method effectively promoted the formation of the VOCs (flavor compounds) in muscle tissues, which was similar to the 100 °C thermal processing treatments. In comparison with the SV-75-0 samples, the composition and content of the VOCs were greatly altered in the scallop samples (SV-75-30) after 30 d of chilled storage, and this was likely due to the growth and reproduction of residual bacteria and the hydrolysis of endogenous enzymes in muscle tissues during long-term storage [38].

The fingerprinting plot (Figure 5B) of the VOCs was generated to visualize the differences between the different scallop samples. In this study, a total of 42 VOCs were determined in the scallop samples, including 13 aldehydes, 6 alcohols, 3 ketones, 2 acetates, 1 amine, and 17 unidentified compounds (Table 2). Herein, aldehydes mainly resulted from the oxidation of unsaturated fatty acids (such as valeraldehyde, heptyl aldehyde, and caprylic aldehyde), triglycerides, and the degradation of several amino acids [39]. Since the sensory threshold of aldehyde compounds is relatively low, they readily react with other substances to advance the characteristic meat flavors [40]. Among the alcohol components, saturated alcohols have high sensory thresholds and provide small contributions to the overall flavor, while unsaturated alcohols have lower sensory thresholds and present mushroom and metal-like aromas. Their contribution to the overall flavor of meat products is significant [41].

Compared with fresh samples, the contents of 3-pentanone, 1-butanol, and 1-octene-3-ol significantly decreased in the cooked scallop samples (CK-0 and SV-75-0), due to the thermal heating treatments. Compared with the CK-0 samples, the contents of ethyl acetate, 2-methylbutanol, 2-pentenal, and acetone were comparatively lower than in the SV-75-0 samples, which suggests the fruity aromas, fishy smell, and fat aroma were reduced in the SV-cooked meat samples. In addition, the intensity of nonanal and ethyl acetate in the SV-cooked scallop samples decreased with a prolonged storage period, and the contents of trimethylamine, 2-hexenal, and 2-methyl butanol increased during 30 d of storage. These results indicate that the fruity, nutty, honey, and citrus aromas were weakened, while the fishy and rancid odors developed during storage, and this was due to the deterioration in muscle tissues caused by the specific spoilage bacteria and several endogenous cathepsins [42]. Overall, the changes in VOCs in the SV-75 scallop samples were relatively small during 0 to 15 d of storage, which indicated that the quality of these samples was comparatively acceptable and stable during this time frame. These results were in agreement with the above TVBN, pH, and MDA results.

## 4. Conclusions

The physicochemical quality and volatile flavor profile of scallop muscle cooked at 75 °C for 30 min were investigated during 30 d of chilled storage. The results show that the physicochemical (TVBN content, pH, MDA content, and texture) and VOC contents of scallop samples treated with traditional heating at 100 °C for 10 min (CK), SV cooking at 70 °C for 30 min (SV-70), and SV cooking at 75 °C for 30 min (SV-75) gradually deteriorated with a prolonged storage period. Importantly, the SV cooking treatments significantly improved the water-holding capacity of scallop samples, as indicated by less weight loss, and maintained a higher MP extraction rate, compared with the CK samples. In addition, there were no significant differences between the VOC contents of SV-75 and CK samples after treatment. Here, the SV-75 samples effectively preserved the quality of half-shell scallops and extended their shelf life to 15–20 d of chilled storage, compared with the SV-70 samples. Scallop products processed by SV cooking and marketed under chilled conditions may be a promising choice for “ready to cook” or “ready to eat” seafood, with high nutritional value and flavor.

## Figures and Tables

**Figure 1 foods-11-03928-f001:**
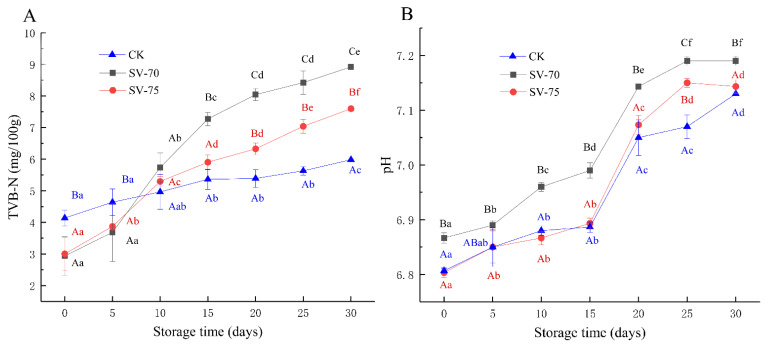
Changes in TVBN content (**A**) and pH value (**B**) of scallop samples with different treatments during 30 d of chilled storage. Different lowercase letters in the same color for the same point plot indicate significant difference (*p* < 0.05), and different uppercase letters in the same storage time indicate significant difference (*p* < 0.05).

**Figure 2 foods-11-03928-f002:**
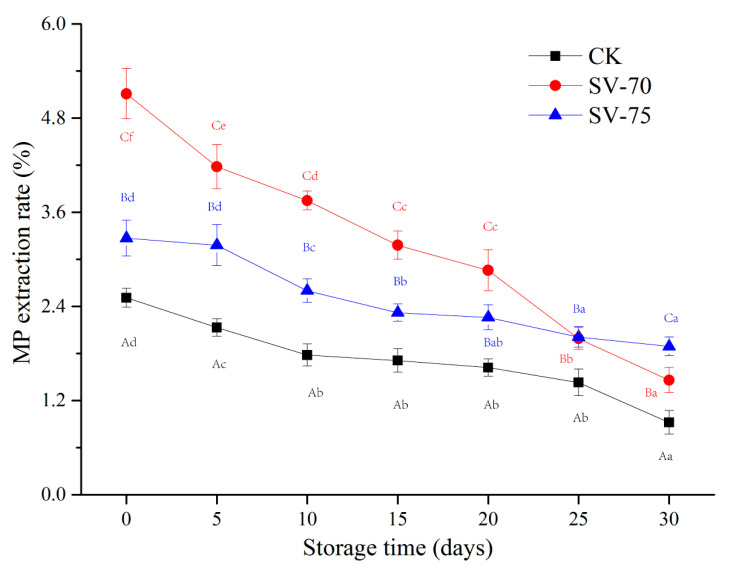
Changes in the MP extraction rate of scallop muscle with different treatments during 30 d of chilled storage. Different lowercase letters in the same color for the same point plot indicate significant difference (*p* < 0.05), and different uppercase letters in the same storage time indicate significant difference (*p* < 0.05).

**Figure 3 foods-11-03928-f003:**
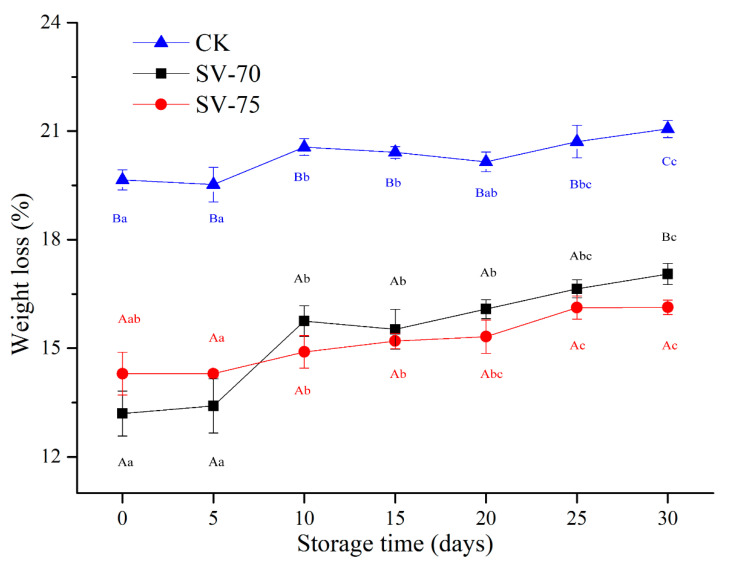
Changes in weight loss of scallop muscle with different treatments during 30 d of chilled storage. Different lowercase letters in the same color for the same point plot indicate significant difference (*p* < 0.05), and different uppercase letters in the same storage time indicate significant difference (*p* < 0.05).

**Figure 4 foods-11-03928-f004:**
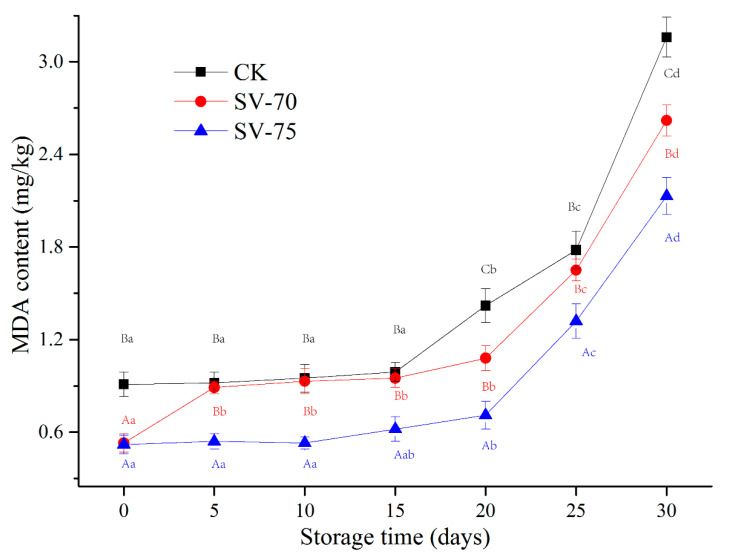
Changes in MDA content of scallop muscle with different treatments during 30 d of chilled storage. Different lowercase letters in the same color for the same point plot indicate significant difference (*p* < 0.05), and different uppercase letters in the same storage time indicate significant difference (*p* < 0.05).

**Figure 5 foods-11-03928-f005:**
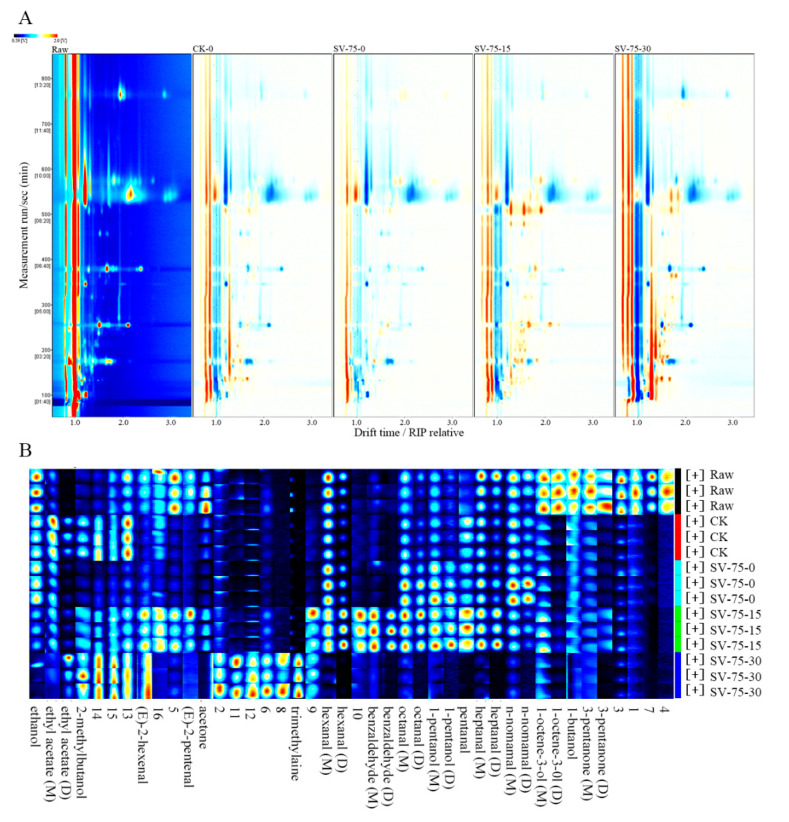
Two-dimensional spectrum (**A**) and fingerprinting plot (**B**) of volatile organic compounds in scallop muscle. Raw, fresh scallops; CK, scallop samples cooked at 100 °C for 10 min and stored for 0 d; SV-75-0, SV-75-15, and SV-75-30, scallop samples cooked at 75 °C for 30 min and stored for 0, 15, and 30 d, respectively.

**Table 1 foods-11-03928-t001:** Changes in springiness and shearing force of scallop muscle with different treatments during 30 d of chilling storage.

Texture Property	Group	Chilled Storage Period						
		0 d	5 d	10 d	15 d	20 d	25 d	30 d
Springiness (mm)	CK	^A^ 1.36 ± 0.02 ^a^	^A^ 1.31 ± 0.03 ^ab^	^A^ 1.28 ± 0.02 ^b^	^A^ 1.26 ± 0.02 ^bc^	^A^ 1.21 ± 0.03 ^cd^	^A^ 1.18 ± 0.01 ^de^	^A^ 1.13 ± 0.01 ^e^
	SV-70	^B^ 1.19 ±0.04 ^a^	^B^ 1.14 ± 0.04 ^ab^	^B^ 1.11± 0.02 ^bc^	^B^ 1.09 ± 0.02 ^bc^	^B^ 1.04 ± 0.02 ^cd^	^B^ 1.01 ± 0.02 ^de^	^B^ 0.96 ± 0.02 ^e^
	SV-75	^A^ 1.29 ± 0.05 ^Ba^	^A^ 1.25 ± 0.01 ^ab^	^A^ 1.23 ± 0.03 ^abc^	^A^ 1.21 ± 0.03 ^bc^	^A^ 1.17 ± 0.01 ^cd^	^A^ 1.15 ± 0.01 ^de^	^A^ 1.10 ± 0.03 ^e^
Shearing force (N)	CK	^A^ 7.79 ± 0.25 ^e^	^A^ 9.23 ± 0.26 ^d^	^A^ 9.80 ± 0.23 ^c^	^A^ 10.04 ± 0.17 ^bc^	^A^ 10.39 ± 0.10 ^b^	^A^ 11.99 ± 0.07 ^a^	^A^ 9.31 ± 0.05 ^d^
	SV-70	^B^ 3.06 ± 0.17 ^e^	^B^ 3.56 ± 0.14 ^cd^	^B^ 3.77 ± 0.14 ^b^	^B^ 4.33 ± 0.28 ^a^	^C^ 3.57 ± 0.16 ^c^	^C^ 3.40 ± 0.23 ^cde^	^C^ 3.11 ±0.14 ^de^
	SV-75	^B^ 3.29 ± 0.33 ^d^	^B^ 3.64 ± 0.24 ^cd^	^B^ 3.89 ± 0.12 ^bc^	^B^ 4.30 ± 0.05 ^ab^	^B^ 4.49 ± 0.10 ^a^	^B^ 4.76 ± 0.29 ^a^	^B^ 3.62 ±0.12 ^cd^

Different uppercase letters in the same column for the same parameter indicate significant difference (*p* < 0.05), and different lowercase letters in the same row indicate significant difference (*p* < 0.05).

**Table 2 foods-11-03928-t002:** The volatile organic compounds (except for the unidentified compounds) in scallop muscle with different treatments.

Compounds	Molecular Formula	Peak Intensity				
		Raw	CK	SV-75-0	SV-75-15	SV-75-30
n-Nonanal (M)	C_9_H_18_O	1356 ± 143 ^b^	1344 ± 70 ^b^	1661 ± 280 ^a^	1146 ± 91 ^b^	581 ± 65 ^c^
n-Nonanal (D)	C_9_H_18_O	232 ± 68 ^ab^	205 ± 18 ^bc^	341 ± 54 ^a^	173 ± 16 ^bc^	69 ± 3 ^d^
Octanal (M)	C_8_H_16_O	868 ± 74 ^b^	985 ± 41 ^b^	1341 ± 140 ^a^	1557 ± 76 ^a^	543 ± 14 ^c^
Octanal (D)	C_8_H_16_O	171 ± 23 ^c^	166 ± 20 ^c^	302 ± 51 ^b^	409 ± 43 ^a^	70 ± 6 ^d^
Benzaldehyde (M)	C_7_H_6_O	449 ± 67 ^bc^	468 ± 32 ^bc^	367 ± 31 ^c^	2008 ± 53 ^a^	570 ± 41 ^b^
Benzaldehyde (D)	C_7_H_6_O	341 ± 40 ^b^	170 ± 15 ^c^	210 ± 41 ^bc^	1399 ± 107 ^a^	195 ± 13 ^bc^
Heptanal (M)	C_7_H_14_O	1176 ± 129 ^b^	902 ± 55 ^b^	991 ± 91 ^b^	1284 ± 91 ^a^	372 ± 19 ^c^
Heptanal (D)	C_7_H_14_O	395 ± 43 ^b^	210 ± 26 ^c^	252 ± 50 ^c^	552 ± 84 ^a^	38 ± 2 ^d^
Hexanal (M)	C_6_H_12_O	1457 ± 64 ^a^	1082 ± 57 ^b^	1431 ± 59 ^a^	1632 ± 25 ^a^	364 ± 28 ^c^
Hexanal (D)	C_6_H_12_O	897 ± 40 ^b^	492 ± 58 ^c^	866 ± 70 ^b^	2268 ± 139 ^a^	92 ± 21 ^d^
2-Pentenal (E)	C_5_H_8_O	197 ± 19 ^a^	48 ± 6 ^b^	77 ± 1 ^b^	217 ± 26 ^a^	37 ± 3 ^b^
Pentanal	C_5_H_10_O	96 ± 13 ^c^	187 ± 26 ^b^	150 ± 10 ^b^	270 ± 10 ^a^	22 ± 2 ^d^
2-Hexenal (E)	C_6_H_10_O	90 ± 11 ^c^	60 ± 5 ^cd^	58 ± 1 ^d^	139 ± 14 ^b^	192 ± 22 ^a^
1-Pentanol (M)	C_5_H_12_O	380 ± 23 ^b^	335 ± 20 ^b^	575 ± 49 ^a^	603 ± 51 ^a^	188 ± 9 ^c^
1-Pentanol (D)	C_5_H_12_O	48 ± 6 ^b^	37 ± 5 ^bc^	86 ± 6 ^a^	103 ± 11 ^a^	23 ± 1 ^c^
Ethanol	C_2_H_6_O	885 ± 94 ^a^	682 ± 48 ^b^	797 ± 22 ^ab^	421 ± 15 ^c^	223 ± 10 ^d^
1-Butanol	C_4_H_10_O	266 ± 21 ^a^	141 ± 10 ^b^	141 ± 9 ^b^	131 ± 5 ^bc^	103 ± 18 ^c^
1-Octene-3-ol (M)	C_8_H_16_O	1877 ± 46 ^a^	1002 ± 45 ^c^	1013 ± 98 ^c^	1466 ± 50 ^b^	1092 ± 57 ^c^
1-Octene-3-ol (D)	C_8_H_16_O	2233 ± 118 ^a^	297 ± 36 ^c^	310 ± 63 ^bc^	467 ± 56 ^b^	287 ± 24 ^c^
2-Methylbutanol	C_5_H_12_O	124 ± 23 ^c^	338 ± 16 ^b^	122 ± 14 ^c^	359 ± 24 ^b^	479 ± 32 ^a^
3-Pentanone (M)	C_5_H_10_O	336 ± 24 ^a^	150 ± 7 ^c^	180 ± 5 ^b^	174 ± 2 ^bc^	60 ± 7 ^d^
3-Pentanone (D)	C_5_H_10_O	418 ± 82 ^a^	69 ± 7 ^c^	61 ± 5 ^c^	174 ± 18 ^b^	25 ± 5 ^d^
Acetone	C_3_H_6_O	3713 ± 68 ^a^	1383 ± 90 ^c^	720 ± 44 ^d^	2271 ± 92 ^b^	2182 ±44 ^b^
Ethyl acetate (M)	C_4_H_8_O_2_	246 ± 52 ^c^	692 ± 72 ^a^	396 ± 54 ^b^	420 ± 30 ^b^	118 ± 17 ^d^
Ethyl acetate (D)	C_4_H_8_O_2_	41 ± 7 ^e^	354 ± 27 ^b^	87 ± 19 ^d^	157 ± 25 ^c^	577 ± 60 ^a^

Different letters in the same row indicate significant differences at *p* < 0.05.

## Data Availability

The data presented in this study are available on request from the corresponding author.

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
