# Peer review of "Effects of Sous Vide Cooking on the Physicochemical and Volatile Flavor Properties of Half-Shell Scallop (Chlamys farreri) during Chilled Storage"

_foods, 2022, doi:10.3390/foods11233928_

Round 1

Reviewer 1 Report

The presented work is difficult to review due to the lack of important information about the research material and research methods. The samples weighed 60-70 g. Such a mass is too small to be able to perform all tests on the same samples. So I assume that the research was carried out on different samples. They could have come from different sources (from different sellers). Moreover, they could be collected at different times (no information when the samples were taken).There is no information on the total number of samples and the number of samples in the groups.

The description of the research methods should be more detailed. The descriptions of the research methods lack information about the mass and size of the analyzed samples. Not all descriptions of procedures are accurately described (e.g. lines 108-110 - homogenate preparation, 116-118 - buffer volume and homogenization rate).

In some parts of the text, references to literature are missing (for example, lines: 42-43, 47-49, 96-98, 108-114, 124). The authors report that the differences between the group means were significant, but do not provide information on the significance level.

Due to the presented methodological comments, I am not able to assess the real value of the presented research results.

Author Response

Thank you for the comments on our manuscript titled "Effects of sous vide cooking on the physicochemical and volatile flavor properties of half-shell scallop (Chlamys farreri) during chilled storage" (No. foods-1979081).

These comments provided important guidance for our following research. We addressed the reviewers’ comments to the best of our abilities and revised the text to meet the requirements. We hope this meets your requirements for a publication. We marked the revisions in red in the manuscript. The main comments and specific responses are detailed below:

  1. The presented work is difficult to review due to the lack of important information about the research material and research methods. The samples weighed 60-70 g. Such a mass is too small to be able to perform all tests on the same samples. So I assume that the research was carried out on different samples. They could have come from different sources (from different sellers). Moreover, they could be collected at different times (no information when the samples were taken). There is no information on the total number of samples and the number of samples in the groups.

Response: Detailed information about the material and methods were added in the revised text, including numbers, weight, length, height, and harvested time. Please check the revisions and hope to meet your criterion.

  1. The description of the research methods should be more detailed. The descriptions of the research methods lack information about the mass and size of the analyzed samples. Not all descriptions of procedures are accurately described (e.g. lines 108-110 - homogenate preparation, 116-118 - buffer volume and homogenization rate).

Response: Detailed information about the material and methods were added, including numbers, weight, length, height, and harvested time. Homogenate preparation, buffer volume, and homogenization rate were also added in the revised text. Please check the revisions and hope to meet your criterion.

  1. In some parts of the text, references to literature are missing (for example, lines: 42-43, 47- 49, 96-98, 108-114, 124). The authors report that the differences between the group means were significant, but do not provide information on the significance level.

Response:  The supported references were added to the text. And authors added significance analysis to the Figures. Please reviewer check the improved Figures.

  1. Due to the presented methodological comments, I am not able to assess the real value of the presented research results.

Response: Detailed information about the material and methods were added, including numbers, weight, length, height, and harvested time. Homogenate preparation, buffer volume, and homogenization rate were also added in the revised text. Please check the revisions and hope to meet your criterion.

Reviewer 2 Report

The manuscript has investigated the effects of sous vide cooking on the physicochemical and volatile flavor properties of half-shell scallops during cold storage. The manuscript is well-written and interesting.

Comments: 

Line 193: drip loss

Line 197: drip loss

Figures: statistical analysis is not performed. I suggest using tables to be able to show the differences between treatments and storage times.

Table 1: It is stated that “Different uppercase letters in the same column indicate significantly different” however two parameters (springiness and shearing force) are shown in one column. Please write the data in separate tables or add space or draw a line between these two parameters and explain that different letters for each parameter show significant differences.  

Line 276: higher  

Author Response

Thank you for the comments on our manuscript titled "Effects of sous vide cooking on the physicochemical and volatile flavor properties of half-shell scallop (Chlamys farreri) during chilled storage" (No. foods-1979081).

These comments provided important guidance for our following research. We addressed the reviewers’ comments to the best of our abilities and revised the text to meet the requirements. We hope this meets your requirements for a publication. We marked the revisions in red in the manuscript. The main comments and specific responses are detailed below:

  1. Line 193: drip loss; Line 197: drip loss

Response: They were corrected according to the revisions.

  1. Figures: statistical analysis is not performed. I suggest using tables to be able to show the differences between treatments and storage times.

Response: Thank you for the suggestions. That’s all right, table results could present differences between treatments and storage times, but it can not show the trend during storage. The authors added significance analysis to the Figures. Please reviewer check the improved Figures and hope meeting with your criterion.

  1. Table 1: It is stated that “Different uppercase letters in the same column indicate significantly different” however two parameters (springiness and shearing force) are shown in one column. Please write the data in separate tables or add space or draw a line between these two parameters and explain that different letters for each parameter show significant differences.

Response: A line between two parameters has been drawn in the table, and the sentence was corrected as "Different uppercase letters in the same column for the same parameter indicate significantly different".

  1. Line 276: higher

Response: It was corrected according to the revision.

Reviewer 3 Report

The paper describes the main findings on the effects of SV cooking on the physicochemical and volatile flavour properties of half-shell scallops during chilled storage. Although the study methods are well-defined, some major points need to be clarified before considering publication. While I can see the conceptual frame on which the Authors have built their work, I have problems seeing the originality of this paper and seeing any breakthrough results. The main point here is if SV cooking was effective to prolong the shelf-life of the scallops or not and how important were the differences found on VOC content. Without microbiological analysis these questions cannot be answered. In summary, I understand that the Authors did a lot of work, but in my opinion unfortunately the paper contains few novelty,  present version needs revision, and comments are attached.

Author Response

Thank you for the comments on our manuscript titled "Effects of sous vide cooking on the physicochemical and volatile flavor properties of half-shell scallop (Chlamys farreri) during chilled storage" (No. foods-1979081).

These comments provided important guidance for our following research. We addressed the reviewers’ comments to the best of our abilities and revised the text to meet the requirements. We hope this meets your requirements for a publication. We marked the revisions in red in the manuscript. The main comments and specific responses are detailed below:

  1. The paper describes the main findings on the effects of SV cooking on the physicochemical and volatile flavour properties of half-shell scallops during chilled storage. Although the study methods are well-defined, some major points need to be clarified before considering publication. While I can see the conceptual frame on which the Authors have built their work, I have problems seeing the originality of this paper and seeing any breakthrough results. The main point here is if SV cooking was effective to prolong the shelf-life of the scallops or not and how important were the differences found on VOC content. Without microbiological analysis these questions cannot be answered. In summary, I understand that the Authors did a lot of work, but in my opinion unfortunately the paper contains few novelty to be published in its present version.

Response: Thank you for the comments about our manuscript. The authors agree with the reviewer’s concerns about the microbiological analysis. The microbiological test in the half-shell scallops during storage is still in progress by using high-throughput sequencing procedures, including a 16S rRNA (for bacteria) analysis and an 18S rDNA (for mould) analysis combined with shelf-life analysis. The current manuscript, as we can see, has included 5 figures and 1 table, so it’s not easy to add more results to the text. In the case of microbiological analysis, a new manuscript will be prepared and submitted to clarify the microbiological analysis.

As the reviewer considered, the current study seems to lack novelty, as SV cooking is a known cooking method. While, as we know, the application of SV cooking in shellfish products is still a prime research objective according to publications on the Web of Science system. So, the current study was performed to evaluate the SV cooking effects on the physicochemical properties of half-shell scallops during storage. Please reviewer considers this possibility and hope to meet with your approve.

  1. Lines 35-37: Please consider adding a reference on the microbiological quality of fresh bivalve molluscs such as mussels, scallops etc. Parlapani, F. F., et al. "HRM analysis as a tool to facilitate identification of bacteria from mussels during storage at 4 C." Food microbiology 85 (2020): 103304.

Response: Two references including the one you recommended have been added. The sentence was also improved according to your revisions.

  1. Lines 156-163: This is general, well-known information. Could the authors please make a more thorough discussion regarding the changes to the TVBN content?

Response: These sentences were improved. According to current literature, changes in TVBN content are mainly correlated with the activity of endogenous enzymes, spoilage microorganisms, and several oxidation reactions. In-depth mechanisms are insufficient, which also provide a good research direction for our team. Thank you for the suggestions.

  1. Lines 163-165: It would be better to discuss/compare your results with other seafood products having a holistic picture. Please, find a relative reference below: Kritikos, Athanasios, et al. "Volatilome of chill-stored European seabass (Dicentrarchus labrax) fillets and Atlantic salmon (Salmo salar) slices under modified atmosphere packaging." Molecules 25.8 (2020): 1981.

Response: This reference you recommended has been added to the text, and discussions compared with European seabass and Atlantic salmon have been included in the content.

  1. Lines 167-167: Have you done any microbiological analysis to prove this? It would be better to avoid any generalizations.

Response: Authors agree with the reviewer’s concerns on the microbiological analysis. The microbiological test in the half-shell scallops during storage is still in progress by using high-throughput sequencing procedures, including a 16S rRNA (for bacteria) analysis and an 18S rDNA (for mould) analysis combined with shelf-life analysis. In the current manuscript, as we can see, has included 5 figures and 1 table, so it’s not easy to add more results to the text. In the case of microbiological analysis, a new manuscript will be prepared and submitted to clarify the microbiological analysis.

  1. Lines 170-171: Please revise. Of course, SV will reduce the number of bacteria as it is a heat treatment.

Response: This sentence was added to the text according to your suggestions.

  1. Line 171: Please, mention the temperature of the SV cooking here.

Response: The temperatures (60°C and 65°C) have been added to the text.

  1. Line 173: Please, specify. Which results?

Response: This sentence was revised.

  1. Figure 1. Please, provide statistical analysis.

Response: Authors added significance analysis in the Figures.

  1. Lines 182-186: Please, revise. The authors have not tested that. You can use examples from the published literature.

Response: This sentence was revised, and a supported reference was also added.

  1. Lines 195-196: This is true, and it can be seen that the results are probably statistically significant; however, why no statistical analysis was performed? I suggest the authors perform a statistical analysis and show at least if the SV-70 and SV-75 had any significant weight loss differences.

Response: Thank you for the suggestions. The authors added significance analysis to the Figures.

  1. Lines 199-200: Please, perform statistical analysis for all your results.

Response: Authors added significance analysis in the Figures.

  1. Line 204: Please, revise this sentence. Does not make sense.

Response: This sentence was revised according to your suggestions.

  1. Table 1. In table 1, please consider moving the uppercase letters on the left side of each value and the lowercase on the right to be more straightforward for the reader.

Response: According to your suggestions, the uppercase letters are on the left side of each value.

  1. Lines 236-237: Please, add a reference.

Response: The reference was added to the text.

  1. Lines 259-261: Could the authors please mention they observed this effect? Were the results significantly different?

Response: Authors added significance analysis in the Figures.

  1. Line 287: Please, revise to "this could be attributed"!

Response: Line 211 now read "this could be attributed to ...".

  1. Lines 296-297: Please, mention why you have chosen only the CV-75 samples to investigate VOC changes.

Response: The reasons were added in the text according to the revisions.

  1. Lines 308-310: Please, add reference(s).

Response: A reference was added.

  1. Line 347: Please, revise. The volatile flavour profile was tested only for the SV-75 scallops.

Response: This sentence was revised.

  1. Lines 63-64: Can this work be relevant and helpful to other marine bivalve molluscs, such as mussels? Is there really a large-scale production and preparation of RTE-cooked only half shells? The final aim of this work looks a bit weak, and I cannot find the novelty.

Response: Thank you for the important suggestions. The information was added in the Introduction section according to the revisions to improve the significance.

  1. Line 203-204: This is a major issue that needs to be revised throughout the manuscript. The authors have not performed any microbiological analysis and need to stop stating the microbiological changes and their effects in a way like they did an analysis. This is an important limitation of the current study, and if the authors cannot provide a microbiological analysis must mention this.

Response: The microbiological test in the half-shell scallops during storage is still in progress by using high-throughput sequencing procedures, including a 16S rRNA (for bacteria) analysis and an 18S rDNA (for mould) analysis combined with shelf-life analysis. In the current manuscript, as we can see, has included 5 figures and 1 table, so it’s not easy to add more results to the text. In the case of microbiological analysis, a new manuscript will be prepared and submitted to clarify the microbiological analysis.

Anyway, the authors completely agree with the reviewer’s considerations that the microbiological test is a shortcoming of the current submission, which will be discussed in our subsequent reports (bacteria and mould). The statement on the microbiological analysis was added in the TVBN section. Please reviewer check the revisions and hope that these considerations will meet with approval.

  1. Lines 263-265: Even if the authors did not perform any microbiological analysis, it is imperative to add a few lines and introduce the most common shelf-life of fresh, cooked, and treated SV scallops. This will help to understand if the reported results of the current research are important or not for the seafood industry and the consumers.

Response: Thank you for your valuable suggestions. According to the literature, there are few studies on SV-treated scallops published on the Web of Science. The authors discussed several related references in the text. Please check the added content and hope to meet with your approval.

  1. Line 269: Sub-section 3.5. I suggest the authors move this section after the "TVBN and pH" analysis.

Response: Previous sub-section 3.5. was moved after the "TVBN and pH" analysis.

  1. Lines 288-291: This is unclear. Please, mention the Day and provide statistical analysis to prove the results have statistically significant differences.

Response: This sentence was revised, and significance analysis was also added. in the Figures

  1. Line 357: The authors cannot state that, as I have explained with my comments throughout the manuscript.

Response: This sentence was revised according to the suggestions.

Round 2

Reviewer 3 Report

Comments

The paper has been improved significantly. Overall, the article is well-structured and appropriately written. A few typos and minor changes need to be addressed before publication in foods.

Minor Comments

 Figures 1, 2, 3, and 4. Please, correct the figures’ legends. Revise the words “significantly different” to “significant difference”.

Figures 1, 2, 3, and 4. Please, improve the quality of the Figures as there is an obvious shadow on each point.

Table 1. Please, revise the words “significantly different” to “significant difference”.

Lines 252-253. Please, be more specific. Your results were consistent with which previous results?

Author Response

Thank you for the comments on our manuscript titled "Effects of sous vide cooking on the physicochemical and volatile flavor properties of half-shell scallop (Chlamys farreri) during chilled storage" (No. foods-1979081).

These comments provided important guidance for our following research. We addressed the reviewers’ comments to the best of our abilities and revised the text to meet the requirements. We hope this meets your requirements for a publication. We marked the revisions in red in the manuscript. The main comments and specific responses are detailed below:

The paper has been improved significantly. Overall, the article is well-structured and appropriately written. A few typos and minor changes need to be addressed before publication in foods.

Response: Thank you for the comments about our manuscript.

Figures 1, 2, 3, and 4. Please, correct the figures’ legends. Revise the words “significantly different” to “significant difference”.

Response: The figures’ legends have been revised.

Figures 1, 2, 3, and 4. Please, improve the quality of the Figures as there is an obvious shadow on each point.

Response: The figures have been replaced with a higher quality.

Table 1. Please, revise the words “significantly different” to “significant difference”.

Response: “significantly different” has been revised as “significant difference”.

Lines 252-253. Please, be more specific. Your results were consistent with which previous results?

Response: This sentence now reads “This is consistent with the results of previous TVBN and pH measurements, which also indicate the 75°C treatment is better for maintaining the scallop muscle quality than the 70°C treatment”.